# Effect of Different Cooking Treatments on the Residual Level of Nitrite and Nitrate in Processed Meat Products and Margin of Safety (MoS) Assessment

**DOI:** 10.3390/foods12040869

**Published:** 2023-02-17

**Authors:** Marco Iammarino, Giovanna Berardi, Igor Tomasevic, Valeria Nardelli

**Affiliations:** 1Department of Chemistry, Istituto Zooprofilattico Sperimentale della Puglia e della Basilicata, 71121 Foggia, Italy; 2German Institute of Food Technologies (DIL), 49610 Quackenbruck, Germany

**Keywords:** cured meats, Margin of Safety, meat cooking, nitrate, nitrite, risk exposure, total diet study

## Abstract

Nitrite and nitrate are well-known food additives used in cured meats and linked to different food safety concerns. However, no study about the possible effect of cooking treatment on the residual level of these compounds before consumption is available. In this work, 60 samples of meat products were analyzed in order to evaluate the variation in residual nitrite and nitrate level after baking, grilling and boiling. The analyses by ion chromatography demonstrated that meat cooking leads to a decrease in nitrite and an increase in nitrate residual levels in the final products. Meat boiling caused an overall decrease in two additives’ concentration, while baking and particularly grilling caused an increase in nitrate and, in some cases, nitrite as well. Some regulatory aspects were also considered, such as the possibility of revising the legal limit of nitrate from the actual 150 mg kg^−1^ to a more cautious 100 mg kg^−1^. Indeed, several meat samples (bacon and swine fresh sausage) resulted in a higher nitrate concentration than the legal limit after cooking by grilling (eleven samples) or baking (five samples). Finally, the Margin of Safety evaluation demonstrated a good level of food safety, all values being higher than the protective threshold of 100.

## 1. Introduction

Nitrite (NO_2_^−^) and nitrate (NO_3_^−^) are two ions involved in the nitrogen cycle and, being present in the environment also as a consequence of the widespread use of nitrogen fertilizers in agriculture, they can be found in water and some types of food, especially vegetables. Leafy vegetables mainly contribute to the overall intake of nitrate in the diet, due to the very high accumulation capability of the leaves, up to 2978.1, 5101.0, 5834.9 and 7311.2 mg kg^−^^1^ in spinach, lettuce, chard and wild rocket, respectively. In some cases, nitrite can also be detected in high amounts, such as 197.5, 66.5, 131.6 and 219.5 mg kg^−^^1^ in spinach, lettuce, chard and rucola, respectively [1,2,3]. Moreover, their sodium and potassium salts, classified in Europe with food additive codes from E249 to E252, are widely used in food processing for the treatment of meats and some types of cheese and seafood [4,5]. Depending on the food type, these additives can exercise different functions, such as preventing bacterial growth (especially the high-concern *Clostridium botulinum*), improving pink coloration and flavor [5,6,7]. As an example, a limit equal to 150 mg kg^−^^1^ (expressed as NaNO_2_ or NaNO_3_) was established for “Non-heat–treated meat products”. Other limits, established for some traditionally cured meat products, can vary from 10 to 300 mg kg^−^^1^ and from 50 to 180 mg kg^−^^1^ for nitrate and nitrite addition, respectively [4]. Lastly, “natural” levels of nitrate can also be found in foodstuffs with no added food additives [2,8].

It is well known that these two compounds, especially nitrite, can exercise some harmful effects on humans, such as the formation of *N*-Nitrosamines (classified as carcinogenic or possibly carcinogenic) after reaction with secondary and tertiary amines [9,10]; the “Methemoglobinemia” disease, that is, the oxidation of hemoglobin to methemoglobin, which cannot transport oxygen to tissues; and several other adverse reactions in susceptible people [11,12]. The maximum admissible levels of nitrite and nitrate in food, drinking water and animal feed have been regulated worldwide [4,7,9,13,14,15,16,17] and specific Admissible Daily Intakes (ADIs), equal to 0.06, 0.07 and 3.7 mg kg body weight (b.w.)^−^^1^ per day of sodium nitrite, potassium nitrite and sodium/potassium nitrate, respectively, have been established.

It is also worthy of mentioning that recent studies seem to demonstrate that high intakes of nitrate can improve cardiometabolic health and exercise performance [18,19]. These positive effects are due to the entero-salivary production of nitric oxide, which is a significant cellular signaling molecule [20,21,22]. Thus, the food safety topic “nitrite-nitrate in food” should be addressed in a more comprehensive view, such as the “Risk-benefit analysis” [23].

In this regard, the “Total Diet Studies” developed worldwide also demonstrated that food processing, including home cooking, can substantially modify the actual intake of many contaminants on consumption [24,25,26]. However, these studies were only focused on food contaminants, due to their higher toxicity, so food additives were substantially overlooked, since few studies are available [27,28].

This work is conducted in this context. It represents the first report focused on the variation in nitrite and nitrate levels in most consumed types of cured meats, after different types of cooking treatment. The analytical determinations were obtained by means of ion chromatography coupled with suppressed conductivity detection.

## 2. Materials and Methods

### 2.1. Sample Collection and Preparation Procedure for IC Analysis

Regarding sample collection, special attention was devoted to widely-consumed meat products [29] containing nitrite and nitrate (single or in combination) as food preservatives. In this sense, the product label was taken into consideration when choosing the products. The products were collected, following the randomness principle, from local stores during the period 2021–2022. The following products containing both nitrite and nitrate (E250–E252) were purchased: swine wurstel (1 sample), bacon (12 samples) and fresh swine sausage packed under vacuum (15 samples); 32 samples contained only nitrite (E250): chicken wurstel (8 samples), chicken/turkey wurstel (7 samples), swine wurstel (14 samples) and bacon (3 samples). The full list of collected samples, together with the indication of all food additives declared on the products label, is reported in Table 1. Once arrived in the laboratory, the products were stored at −18 °C (±2 °C) until analysis, since this type of storage does not modify the concentrations of nitrite and nitrate, so this is the procedure routinely used for official control activity.

The whole sample of at least 250 g was subdivided into 4 aliquots (the first 100 g and the others 50 g) and stored at −18 °C in the laboratory until analysis. The first aliquot was analyzed 4 times to assure the homogeneity of both additives in the whole sample. The other aliquots were analyzed after 3 different types of cooking treatment (2 repetitions for each test), using the following procedures: grilling (200 °C, 4–6 min), boiling (100 °C, 10 min) and baking (180 °C, 20 min). Each procedure was carried out making reference to the usual domestic procedures, taking care to obtain a proper level of cooking for each sample. Before and after cooking treatments, the samples were completely homogenized using a blade homogenizer. Then, 2 g (±0.01 g) of each sample was transferred to a 250 mL flask, and 100 mL of ultrapure water was added. The samples were then placed in bain-marie at 70 °C ± (3 °C) for 5 min. After vigorous shaking and cooling, an aliquot of at least 3 mL of the supernatant was passed through syringe filters fit for ion chromatography analysis (0.22-µm, Sartorius AG, Goettingen, Germany). In the case of a concentration outside the calibration range (0.6—300 and 0.8—274 mg kg^−^^1^ for sodium nitrite and sodium nitrate, respectively), a proper dilution with ultrapure water was conducted (usually 1/5). The analyses were carried out in duplicate and the final results were expressed as the mean of two measurements.

### 2.2. Chemicals, Standards and Reagents

Nitrate and nitrite reference material (1000 mg L^−^^1^), certified for ion chromatography, were obtained from Sigma-Aldrich (Stenheim, Germany). Na_2_CO_3_ (>99.5%) was supplied by VWR International s.r.l. (Milan, Italy). Ultrapure water (18.2 MΩ-cm), produced using the Arium^®^ mini essential UV system (Sartorius AG, Goettingen, Germany), was used for preparing both mobile phase samples and standard solutions. The calibration curves of nitrite and nitrate were obtained by injecting the following concentrations of both compounds: 0.01, 0.1, 1.0, 5.0 and 10.0 mg L^−^^1^.

### 2.3. Apparatus and Ion Chromatography Method

A high-pressure ion chromatography system composed of a column compartment set at 20 °C, an SP Single Pump (ICS-6000), a gradient mixer (Dionex GM-4, 2 mm), an injection valve with a 25-µL loop, a Dionex self-regenerating suppressor (ADRS 600, 4 mm) set at the recommended voltage and a DC detector in conductivity mode (Thermo Scientific™ Dionex™ ICS-6000 HPIC™ System, Thermo Fisher Scientific Inc., Waltham, MA, USA) were used for the chromatographic determination of nitrite and nitrate. The analytical column adopted was the IonPac^®^ AS9-HC (250 × 2 mm, particle size: 9 µm) equipped with AG9-HC pre-column (Thermo Fisher Scientific Inc.). The analytical approach used in this study was a well-known chromatographic separation, already standardized for meat products analysis [30]. A solution consisting of 9 mM Na_2_CO_3_, degassed with helium and microfiltered before use, was used as the mobile phase. The isocratic elution of both analytes was accomplished by using a flow rate of 1.0 mL min^−^^1^ and a total run time of 25 min. For data acquisition/processing and instrumentation control, the Chromeleon 7.2.8 software (Thermo Scientific) was used.

This analytical method is routinely used for official food control activity at the Chemistry Department of the Istituto Zooprofilattico Sperimentale della Puglia e della Basilicata (Foggia, Italy). In this regard, the analytical procedure was fully validated, following an in-house validation model developed according to the most representative European Regulations and Guidelines [31,32,33,34,35,36,37] and accredited in Italy as of 2004. This analytical method is regularly submitted to Proficiency tests, and it was also successfully applied during an International interlaboratory study, developed for the standardization of a nitrate-rich vegetable food [19]. The limits of quantification (LoQs) of this method correspond to 0.6 and 0.8 mg kg^−^^1^, expressed as sodium nitrite and sodium nitrate, respectively, in matrix. The method accuracy, estimated in terms of precision and recovery percentage, is characterized by mean CV% equal to 4.1% for nitrite and 4.2% for nitrate, and mean recovery percentage of 98.7% and 98.3% for nitrite and nitrate, respectively. The measurement uncertainty is equal to 9.9% for nitrite and 12.0% for nitrate, and the method robustness studies assured its applicability for the analysis of food types other than meats (i.e., fish and dairy products and vegetables) [1,2,3].

### 2.4. Statistical Analysis

When the concentration was lower than the LoQ (only verified for nitrite), an amount corresponding to LoQ/2 was assigned, using these values for data elaboration. This approach, named “middle-bound”, is in accordance with the indications provided by the Italian Institute of Health in the document “Rapporti ISTISAN 04/15” [38].

The statistical analysis was carried out in order to evaluate how different cooking treatments modify the residual level of nitrite and nitrate in meat products. The comparisons were made in terms of variation percentage, with respect to the raw sample, evaluating the one-way ANOVA and the *t*-test, at confident intervals of 95%, 99% and 99.9% (*p* < 0.05, 0.01, 0.001).

The results obtained during the study were also used for a contribution to risk assessment. Since nitrite and nitrate are not carcinogenic substances, it was not possible to calculate the Margin of Exposure (MoE). Thus, the assessment was elaborated in terms of Margin of Safety (MoS), using 1 as the minimum requirement (bigger is better) and 100 as a protective threshold [39]. The estimation of MoS was obtained as the ADI/estimated exposure dose ratio, under a high-exposure scenario for toddlers considering 12 kg as reference b.w., as established by EFSA [9,15,40]. This choice was to the aim of carefully evaluating the most concerning scenario. The high-exposure scenario was elaborated taking into account, for each meat product type and cooking treatment, the highest concentration detected during the study. With regard to meat products consumption, the reference data were obtained from the INRAN-SCAI 2005-06 Italian National surveys, taking into consideration the mean data [41,42]. This approach, very useful for the estimation of global intake of nitrite and nitrate, was successfully used in some recently published papers [3,29].

## 3. Results and Discussion

### 3.1. General Remarks

The results obtained by analyzing 60 samples of meat products for the determination of nitrite and nitrate, both before and after three different types of cooking treatment, are reported in Table 1 and elaborated on in Figure 1. Some chromatogram examples related to both standard solution and meat samples are shown in Figure 2, Figure 3 and Figure 4. A graphical representation of the mean variations in nitrite and nitrate concentrations is also shown in Figure 5. As a first comment, it is worthy of mentioning that the results obtained by analyzing two repetitions for each test demonstrated good repeatability, with SD values and CV% in the range 0.1–9.7 and 2.5–20.2, respectively.

The first significant result is the different effect of cooking treatments on the residual level of nitrite and nitrate. Indeed, cooking, generally speaking, leads to a decrease in nitrite concentration and an increase in nitrate. As a first comment, this result can be justified considering the natural oxidation of nitrite to nitrate due to high temperatures [27,29,43].

Regarding nitrite, boiling caused a significant decrease in the final amount on consumption (*p* < 0.05), equal to 25.8% and 43.5% in swine fresh sausage and chicken/turkey wurstel samples, respectively. This result is in accordance with the findings obtained by McMahon et al. [27] who analyzed different types of vegetables. Although not significant, a substantial reduction (37.6%) was also registered in swine wurstel samples. No significant modification in the final amount was verified in bacon samples. This finding can be due to low amounts detected in the raw samples (mean concentration: 3.6 mg kg^−^^1^) when compared to the other meat product types (mean concentration: 15.3 mg kg^−^^1^). Most other determinations confirmed that both baking and grilling caused a decrease in nitrite level, in the range [8.8–26.1%] and [14.7–47.9%], respectively (Table 2), with few, not significant, exceptions.

Concerning nitrate, cooking by grilling always leads to a significant increase (*p* < 0.01) in the final concentration, in the range [47.8% (swine wurstel)–94.4% (swine fresh sausage)]. This result is very important, since it represents another concern related to this type of meat cooking, already linked to the formation of well-known carcinogenic compounds such as polycyclic aromatic hydrocarbons, nitrosamines and heterocyclic amines [44,45,46]. Although less significant, cooking by baking also resulted in an increase in nitrate concentration, in the percentage range [36.5% (swine wurstel)–77.4% (bacon)]. Cooking by boiling resulted in a variable effect, probably depending on the possible extraction of nitrate from the matrix due to the high temperature of water. Indeed, this effect is exploited in the usual laboratory extraction step of nitrate from meat matrices [30]. It is worthy of mentioning that the decrease in nitrate level after cooking by boiling is always verified in food where the nitrite level is very low, i.e., vegetables [47], since the increase due to nitrite oxidation is negligible.

Another important result is related to the legal limits and the cooking effect on the final compliance with such reference values. Taking into account both the maximum permitted level set in the European Regulation [4] in the meat products considered in this study (150 mg kg^−1^ as NaNO_3_) and the measurement uncertainty of the method (12%), all raw samples were compliant. After cooking, the increase in nitrate concentrations made several samples “not-complaint”, since the level exceeded 150 mg kg^−1^. In particular and in accordance with the comments above, cooking by grilling resulted in the highest number of samples exceeding the legal limit (11), as subdivided: six bacon (range 185.4–293.8 mg kg^−1^, mean: 229.5 mg kg^−1^) and five swine fresh sausage (range: 176.2–233.6 mg kg^−1^, mean: 210.1 mg kg^−1^). Cooking by baking also caused this significant result, in particular in three bacon samples (range 190.6–253.4 mg kg^−1^, mean: 216.4 mg kg^−1^) and two samples of swine fresh sausage (range: 194.7–206.8 mg kg^−1^, mean: 200.8 mg kg^−1^). In Figure 4, an example of chromatograms is shown, where the increase in nitrate concentration registered after grilling a swine fresh sausage sample is appreciable.

These results remark the importance of “Total Diet Studies” when evaluating the actual intake of food additives added in food submitted to cooking practices before consumption. Indeed, the variation caused by cooking can lead to significant variation in the final amount of food additives in the products (both increase and decrease). These variations substantially modify different food safety aspects, such as the risk exposure assessment and the definition of legal limits. In this regard, the following study of risk exposure can be considered a contribution to the evaluation of such factors.

### 3.2. Margin of Safety (MoS) Assessment

Regarding the contribution to risk exposure (Table 3), no MoS value resulted lower than the protective threshold of 100, confirming an acceptable level of food safety. This is mainly due to the original amount of nitrate and nitrite present in the formulations which resulted, as already commented above, not particularly high.

Regarding nitrite, the lowest mean values of MoS were obtained for raw samples, due to cooking which causes the oxidation of nitrite to nitrate. The higher mean value was registered for grilled meats; however, considering the overall increase in nitrite caused by this cooking treatment (Figure 5), the validity of this comment only assumes a comparison meaning.

Regarding nitrate, the mean values of MoS were in the range 17,014 (baking)–40,954 (boiling). From a food safety point of view, the higher value calculated for the boiling procedure confirms that this type of cooking treatment can be considered the best one when evaluating nitrite and nitrate intake both before and after the cooking of meat products.

### 3.3. Discussion and Regulatory Aspects

The first important remark of this study is related to the different effects obtained cooking meats by boiling when compared to grilling and baking. Indeed, the first type of cooking leads to an overall decrease in both additives, while baking and particularly grilling cause an increase in nitrate and, in some cases, nitrite as well. Considering that nitrite is the most important precursor of nitrosamine formation in meats, and the possible positive correlation between heat treatments and the increase in levels of these toxic contaminants in meats [48], the impact of cooking on the overall level of safety of meats assumes a fundamental importance.

The second remark has a significant impact on the regulatory aspects. In several cases, cooking caused an increase in nitrate concentration up to levels above the legal limit defined in the reference regulation. Within these samples, the highest increase percentage was equal to 125% (grilled bacon), while the mean value was 70.9%. This mean increase percentage is comparable to those calculated, considering all data related to bacon and fresh swine sausage cooked by both grilling and baking, which was 69.8%. Thus, as a possible reference parameter, it is possible to hypothesize that the residual concentration of nitrate in these types of most concerning meat product samples increases by ~70% after cooking by grilling or baking. In this regard, also considering an adequate value of measurement uncertainty, it is possible to suggest a revision of the legal limit of nitrate in bacon and sausage from the actual 150 mg kg^−^^1^ to a more cautious 100 mg kg^−^^1^.

## 4. Conclusions

In this study, 60 samples of widely-consumed meat products were analyzed in order to evaluate the variation in residual nitrite and nitrate concentration after three different types of cooking treatments: baking, grilling and boiling.

The analytical determinations, accomplished by means of quality-assured chromatography methods, revealed that cooking treatment generally leads to a decrease in nitrite and an increase in nitrate residual levels in the final products.

Meat boiling causes a general decrease in both additives’ concentration, while baking and particularly grilling cause an increase in nitrate and, in some cases, also nitrite. This result is significant, also considering the possible increase in nitrosamine levels.

Regarding regulatory aspects, in eleven and five meat samples (bacon and swine fresh sausage) cooked by grilling and baking, respectively, cooking caused an increase in nitrate concentration up to levels above the legal limit defined in Reg. No. 1333/2008/EC. This result, other than confirming the importance of considering food cooking within the “Total Diet Studies”, also suggests a revision of the legal limit of nitrate in bacon and sausage from the actual 150 mg kg^−^^1^ to a more cautious 100 mg kg^−^^1^.

Finally, the risk exposure assessment, evaluated in terms of MoS, resulted in acceptable levels of food safety, since all the values were higher than the protective threshold (100).

## Figures and Tables

**Figure 1 foods-12-00869-f001:**
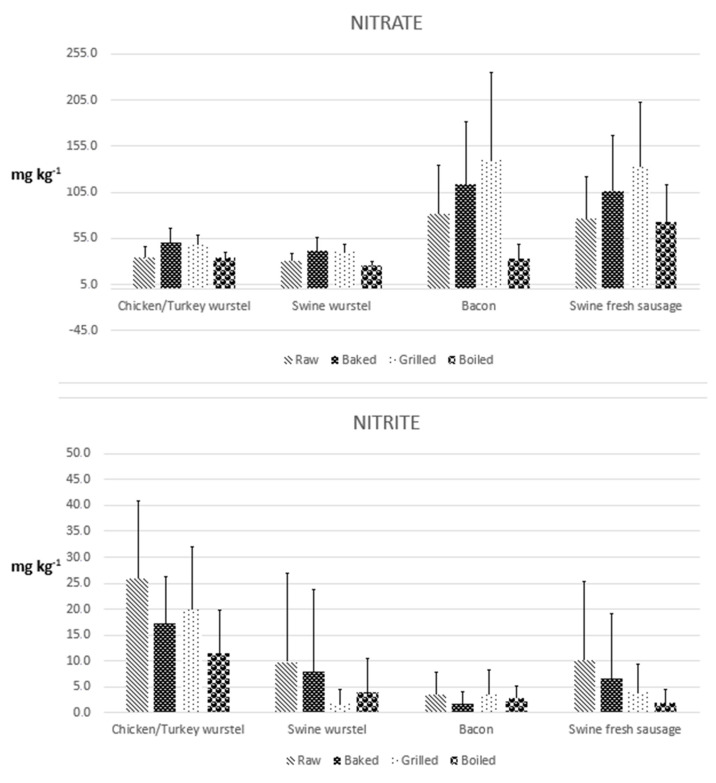
Mean concentrations of nitrite and nitrate quantified analyzing 60 samples of meat products both before and after cooking.

**Figure 2 foods-12-00869-f002:**
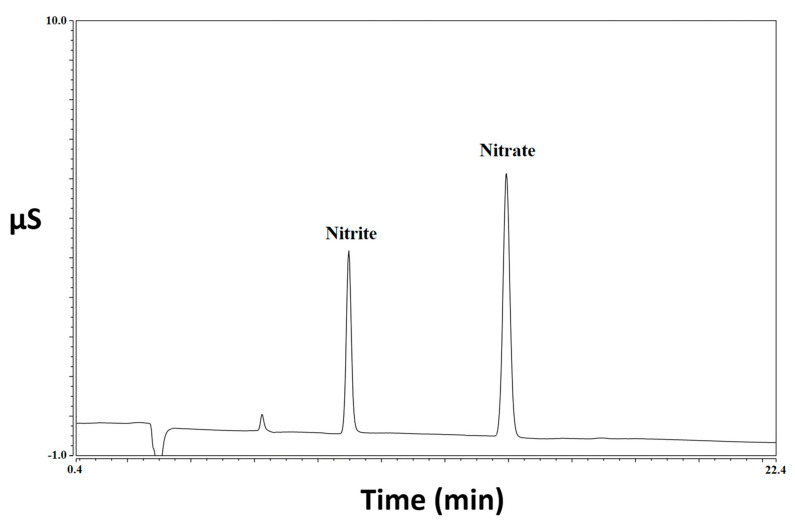
Chromatogram of a standard solution at a concentration of 5.0 mg L^−1^ of nitrite and nitrate.

**Figure 3 foods-12-00869-f003:**
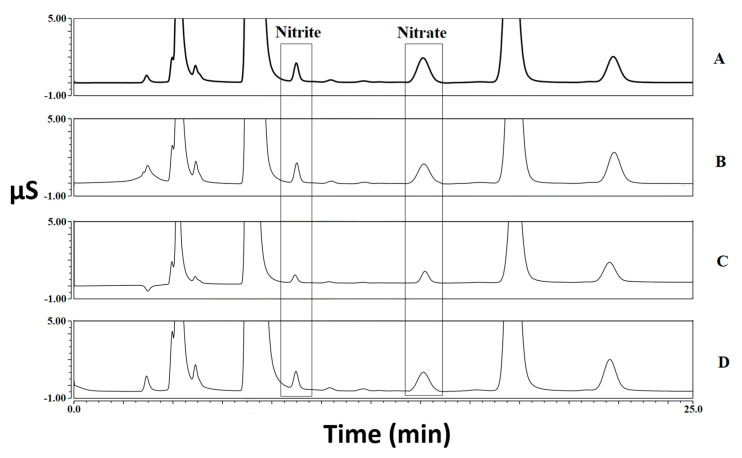
Chromatograms of a chicken wurstel sample both before and after cooking. Nitrite concentrations in the range 21.8 mg kg^−1^ (boiled—**C**)–46.2 mg kg^−1^ (grilled—**B**); Nitrate concentrations in the range 43.0 mg kg^−1^ (boiled—**C**)–89.1 mg kg^−1^ (raw—**A**). **D**: baked sample.

**Figure 4 foods-12-00869-f004:**
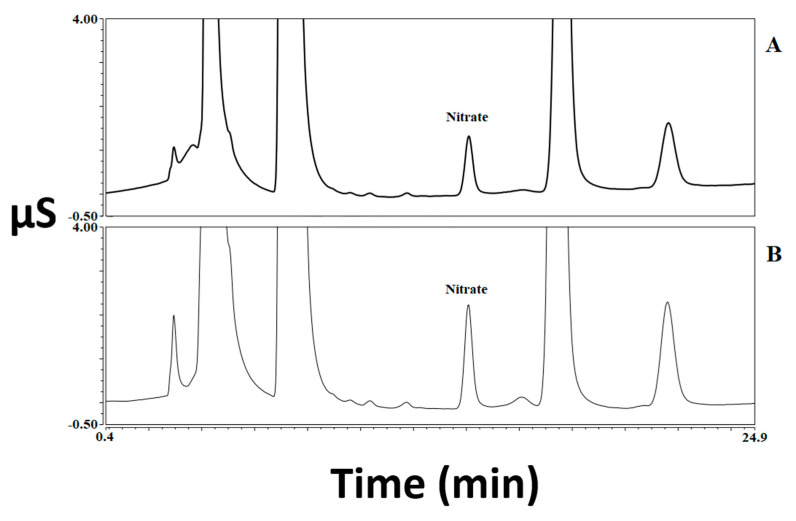
Chromatograms of a swine fresh sausage sample both before and after cooking (extracts diluted 1/5). Nitrate concentrations: 134.6 mg kg^−1^ (raw—**A**); 233.6 mg kg^−1^ (grilled—**B**).

**Figure 5 foods-12-00869-f005:**
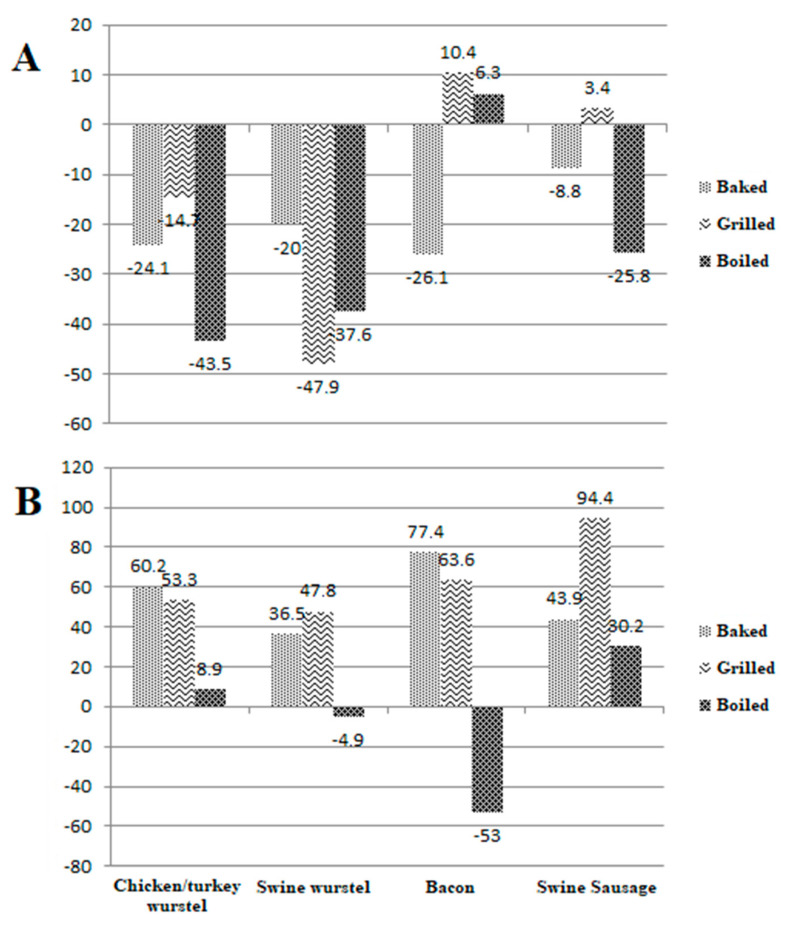
Graphical visualization of mean variations % in nitrite (**A**) and nitrate (**B**) concentrations in 60 samples of meat products after three different types of cooking treatment.

**Table 1 foods-12-00869-t001:** NaNO_2_ and NaNO_3_ concentrations detected in 60 samples of meat products analyzed before and after cooking.

Meat Product	Food Additives Declared on the Label	Nitrite (mg kg^−1^)				Nitrate (mg kg^−1^)			
		Raw	Baked	Grilled	Boiled	Raw	Baked	Grilled	Boiled
Chicken Wurstel	E407, E412, E450, E301, E250	34.5	36.6	38.0	36.6	40.2	45.8	47.6	43.6
Chicken/Turkey Wurstel	E301, E250	65.7	21.8	22.0	13.2	20.2	32.1	39.5	31.1
Chicken Wurstel	E301, E250	38.3	25.1	25.8	14.3	28.0	45.6	49.8	32.9
Chicken/Turkey Wurstel	E301, E250	33.6	16.8	17.0	4.9	20.7	30.9	39.7	29.3
Chicken Wurstel	E301, E250	25.4	17.2	20.0	11.0	35.0	51.1	48.1	33.7
Chicken/Turkey Wurstel	E450, E452, E301, E250	N.D.	N.D.	N.D.	N.D.	32.1	45.1	35.0	33.5
Chicken/Turkey Wurstel	E450, E451, E316, E250	11.0	12.8	15.2	12.8	28.1	47.3	57.3	47.6
Chicken Wurstel	E301, E250	28.7	19.6	29.8	10.4	28.7	107.3	81.5	23.7
Chicken Wurstel	E301, E250	39.5	31.7	46.2	21.8	89.1	75.5	78.5	43.0
Chicken/Turkey Wurstel	E450, E452, E301, E250	13.7	8.9	11.2	6.0	33.6	39.3	34.6	28.2
Chicken/Turkey Wurstel	E301, E250	29.1	23.6	27.7	11.7	31.5	33.7	42.2	23.7
Chicken Wurstel	E301, E250	1.0	1.23	1.3	1.3	22.6	55.4	48.2	33.6
Chicken Wurstel	E301, E250	25.9	17.5	20.5	11.6	34.9	51.0	48.0	34.1
Chicken/Turkey Wurstel	E301, E250	16.1	7.6	2.2	2.1	36.0	51.9	25.0	38.7
Chicken Wurstel	E301, E250	26.3	17.7	21.0	12.2	35.1	50.9	47.9	34.2
Swine Wurstel	E316, E250	18.1	13.3	6.8	2.4	17.6	24.7	40.1	33.3
Swine Wurstel	E301, E250	1.3	1.0	N.D.	N.D.	20.2	30.6	41.6	37.1
Swine Wurstel	E301, E250	9.8	7.9	1.9	3.8	30.7	41.7	41.5	26.0
Swine/Chicken Wurstel	E407, E412, E450, E301, E250	74.4	66.3	N.D.	30.6	56.5	93.6	66.3	33.7
Swine Wurstel	E301, E250	1.7	2.1	N.D.	2.5	26.6	39.5	24.2	18.9
Swine Wurstel	E301, E250	10.0	7.5	1.7	4.2	30.6	41.6	41.8	26.5
Swine Wurstel	E301, E250	1.0	N.D.	1.3	1.1	30.4	81.3	42.9	12.1
Swine Wurstel	E301, E250	N.D.	0.6	N.D.	N.D.	29.2	32.3	47.2	22.6
Swine Wurstel	E301, E621, E450, E452, E250	N.D.	N.D.	N.D.	N.D.	38.1	52.5	40.8	33.7
Swine Wurstel	E301, E250	1.8	N.D.	0.7	1.1	27.9	36.4	35.7	27.5
Swine Wurstel	E331, E262, E301, E250	7.0	6.3	6.3	3.8	36.1	35.8	37.3	29.4
Swine Wurstel	E301, E250	8.6	1.0	0.7	1.0	25.6	26.1	32.2	21.4
Swine Wurstel	E301, E250	1.6	2.2	2.3	2.0	44.8	28.9	49.0	19.2
Swine Wurstel	E301, E250	9.7	8.2	2.1	3.4	30.8	41.8	42.2	25.5
Swine Wurstel	E301, E250, E252	0.9	0.7	N.D.	N.D.	15.7	19.1	45.4	24.7
Bacon	E301, E250, E252	N.D.	0.6	N.D.	N.D.	33.5	68.3	62.9	21.2
Bacon	E301, E250, E252	1.2	N.D.	3.0	1.9	107.2	190.6	185.5	42.1
Bacon	E301, E250	12.4	6.7	4.5	6.6	46.0	31.4	28.7	9.0
Bacon	E301, E250, E252	1.4	N.D.	2.0	1.9	145.5	205.3	204.8	47.7
Bacon	E301, E252, E250	N.D.	N.D.	N.D.	N.D.	128.7	132.5	244.3	28.1
Bacon	E301, E252, E250	N.D.	N.D.	N.D.	0.6	150.8	147.3	293.8	52.4
Bacon	E301, E250, E252	3.3	2.2	3.5	3.1	80.9	114.5	140.0	31.0
Bacon	E250	N.D.	N.D.	N.D.	0.6	116.9	253.4	263.3	58.5
Bacon	E301, E252, E250	N.D.	N.D.	N.D.	N.D.	135.8	150.2	185.4	51.2
Bacon	E250	9.4	8.7	10.0	12.1	21.9	15.7	22.9	12.1
Bacon	E301, E250, E252	3.7	1.8	3.9	2.7	81.6	113.7	150.0	60.2
Bacon	E301, E250, E252	3.8	N.D.	2.1	0.9	13.7	113.4	13.4	14.4
Bacon	E301, E250, E252	3.5	2.0	3.7	2.9	81.3	114.1	140.5	30.6
Bacon	E301, E250, E252	12.9	N.D.	19.1	5.9	24.1	46.1	28.9	16.2
Bacon	E301, E250, E252	N.D.	0.6	N.D.	0.6	52.0	16.0	151.7	14.6
Swine Sausage	E301, E250, E252	8.0	N.D.	15.0	9.3	30.7	19.8	67.6	35.8
Swine Sausage	E301, E250, E252	7.6	12.2	13.7	2.1	45.7	53.3	104.1	33.7
Swine Sausage	E301, E250, E252	6.8	9.3	12.9	6.5	44.5	46.9	73.3	34.2
Swine Sausage	E301, E250, E252	1.8	2.5	2.5	2.7	63.8	105.9	137.1	56.8
Swine Sausage	E301, E250, E252	N.D.	N.D.	0.6	0.6	135.9	206.8	230.6	105.2
Swine Sausage	E301, E250, E252	N.D.	N.D.	0.6	N.D.	134.6	194.7	233.6	95.2
Swine Sausage	E300, E301, E250, E252	48.6	N.D.	N.D.	N.D.	5.8	11.1	18.1	22.2
Swine Sausage	E300, E301, E250, E252	46.7	49.4	N.D.	N.D.	5.3	13.2	13.7	18.9
Swine Sausage	E300, E301, E250, E252	9.6	7.6	3.9	2.0	75.2	106.6	134.5	73.9
Swine Sausage	E300, E301, E250, E252	N.D.	N.D.	N.D.	N.D.	108.4	150.2	157.3	92.4
Swine Sausage	E300, E301, E250, E252	10.5	7.2	4.0	1.7	76.0	105.5	135.5	73.0
Swine Sausage	E301, E331, E250, E252	N.D.	N.D.	N.D.	N.D.	121.2	162.0	195.6	138.7
Swine Sausage	E301, E331, E250, E252	10.1	7.4	4.1	1.9	76.6	106.1	134.0	73.4
Swine Sausage	E301, E331, E250, E252	N.D.	N.D.	N.D.	N.D.	120.5	159.0	200.1	146.1
Swine Sausage	E301, E331, E250, E252	N.D.	N.D.	N.D.	N.D.	103.4	150.7	176.2	102.3

N.D. = Not detected (<0.6 mg kg^−1^).

**Table 2 foods-12-00869-t002:** Mean variation percentages of nitrite and nitrate concentrations after different types of cooking treatment.

Sample Type	N° of Samples Analyzed	N° of Brands Analyzed	Food Preservatives Declared on the Label	N° of Samples with [NO_2_^−^] < LoQ ^a^	[NaNO_2_] (mg kg^−1^) [Range] (Mean) (Raw Sample)	[NaNO_3_] (mg kg^−1^) [Range] (Mean) (Raw Sample)	Mean Variation % [NaNO_2_] after Baking	Mean Variation % [NaNO_2_] after Grilling	Mean Variation % [NaNO_2_] after Boiling	Mean Variation % [NaNO_3_] after Baking	Mean Variation % [NaNO_3_] after Grilling	Mean Variation % [NaNO_3_] after Boiling
Chicken/Turkey Wurstel	15	10	E250	1	[1.0–65.7] (25.9)	[20.2–89.1] (34.4)	−24.1	−14.7	−43.5*	+60.2 **	+53.3 **	+8.9
Swine Wurstel	15	10	E250	2	[0.9–74.4] (9.8)	[15.7–56.5] (30.7)	−20.0	−47.9	−37.6	+36.5	+47.8	−4.9
Bacon	15	10	E250 + E252 (10) E250 (5)	6	[1.2–12.9] (3.6)	[13.7–150.8] (81.3)	−26.1	+10.4	+6.3	+77.4	+63.6 **	−53.0 ***
Swine Fresh Sausage	15	3	E250	6	[1.8–48.6] (10.1)	[5.3–135.9] (76.5)	−8.8	+3.4	−25.8 *	+43.9 *	+94.4 **	+30.2

^a^ LoQ = 0.06 mg kg^−1^ of NaNO_2_; Variation % statistically significant (* *p* < 0.05; ** *p* < 0.01; *** *p* < 0.001). The concentrations increased after cooking are reported in bold.

**Table 3 foods-12-00869-t003:** Margin of Safety (MoS) for meat products both before and after cooking (in brackets: ADI%).

		Raw	Baked	Grilled	Boiled
Nitrite	Chicken wurstel	545 (0.18)	986 (0.10)	774 (0.13)	986
	Swine wurstel	503 (0.20)	567 (0.18)	5538 (0.02)	1229
	Bacon	2927 (0.03)	4337 (0.02)	1962 (0.05)	3117
	Swine fresh sausage	771 (0.13)	760 (0.13)	2509 (0.04)	4045
	Mean MoS	1186 (0.14)	1662 (0.11)	2696 (0.06)	2344
Nitrate	Chicken wurstel	24,944 (0.004)	20,690 (0.005)	27,239 (0.004)	46,737
	Swine wurstel	44,400 (0.002)	24,804 (0.004)	34,961 (0.003)	62,535
	Bacon	15,417 (0.006)	11,327 (0.009)	7914 (0.013)	38,676
	Swine fresh sausage	17,103 (0.006)	11,235 (0.009)	9951 (0.010)	15,868
	Mean MoS	25,466 (0.005)	17,014 (0.007)	20,016 (0.008)	40,954

Mean consumptions of meat products available from: INRAN-SCAI 2005-06. https://www.crea.gov.it/documents/59764/0/appendice_6b5_carne.pdf/6a9412b8-cc45-20c9-7e59-bf6a68dab570?t=1550827578904 accessed on 14 January 2023; % ADI calculated using 12 kg as reference body weight.

## Data Availability

The data presented in this study are available on request from the corresponding author.

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
