# Peer review of "Effect of Different Cooking Treatments on the Residual Level of Nitrite and Nitrate in Processed Meat Products and Margin of Safety (MoS) Assessment"

_foods, 2023, doi:10.3390/foods12040869_

Round 1

Reviewer 1 Report

I have been given this manuscript, "Effect of Different Cooking Treatments on the Residual Level of Nitrite and Nitrate in Processed Meat Products and Margin of Safety (MoS) Assessment" (FOODS-2211457) in order to conduct the review.

From the perspective of food safety control, this study is meaningful. The results of this study may help in the improvement of information regarding the effect of different cooking treatments on the residual level of nitrite and nitrate in processed meat products. It is well presented and discussed. My recommendation is minor revision. I have some suggestions and comments, as follows:

Line 47: Are the ADI values described valid for people of any age and sex? If not, detail it.

Line 48: Replace mg/kg with mg kg-1

Line 95: Add to the text the dilution range that was needed to analyze the samples

Line 97: If each type of cooking treatment was performed in duplicate as well as the sample analyses, the expanded error should be considered. Therefore, I recommend adding the standard deviation values (SD) relative to the concentrations shown in Table 1.

Line 133: Replace “ad” with “as”

Line 132 and 141: Normalize LoD or LOD throughout the manuscript

Table 1: Superscript letters or highlights that indicate statistically significant differences in the concentration of nitrate or nitrite in samples submitted to different cooking treatments would make the reading of the table clearer.

Author Response

The authors would like to thank the reviewer for his effort in improving the scientific impact of the Paper. The manuscript has been revised, according to reviewer’s suggestions, editing corrections and rewording the text where necessary. Please note that the references to lines are referred to the pdf version of the revised article.

Reply to Reviewer 1

I have been given this manuscript, "Effect of Different Cooking Treatments on the Residual Level of Nitrite and Nitrate in Processed Meat Products and Margin of Safety (MoS) Assessment" (FOODS-2211457) in order to conduct the review.

From the perspective of food safety control, this study is meaningful. The results of this study may help in the improvement of information regarding the effect of different cooking treatments on the residual level of nitrite and nitrate in processed meat products. It is well presented and discussed. My recommendation is minor revision. I have some suggestions and comments, as follows:

Line 47: Are the ADI values described valid for people of any age and sex? If not, detail it.

Response: Yes, the ADI values are valid for any age and sex. This is why no detail was added. Thanks for your comment.

Line 48: Replace mg/kg with mg kg-1

Response: Following the reviewer’s remark, the unit was corrected (line 54)

Line 95: Add to the text the dilution range that was needed to analyze the samples

Response: According to the referee’s comment, the usual dilution has been specified (line 105).

Line 97: If each type of cooking treatment was performed in duplicate as well as the sample analyses, the expanded error should be considered. Therefore, I recommend adding the standard deviation values (SD) relative to the concentrations shown in Table 1.

Response: Thanks for your comment. As specified at lines 92-93, the cooking treatment was performed in duplicate. Taking into account the complexity of Table 1, it is very difficult adding all SD values. Consequently, a sentence describing the SD values and the CV%, useful to appraise the overall repeatability of the obtained data, has been added at lines 186-188.

Line 133: Replace “ad” with “as”

Response: Corrected (line 144). Thanks for your careful work.

Line 132 and 141: Normalize LoD or LOD throughout the manuscript

Response: Corrected (lines 154-155 and table 2).

Table 1: Superscript letters or highlights that indicate statistically significant differences in the concentration of nitrate or nitrite in samples submitted to different cooking treatments would make the reading of the table clearer.

Response: Thanks for your comment. Given the low number of data related to each test (n=2) it was not possible to elaborate any statistical analysis in Table 1. The statistical analysis was elaborated relating to each meat products/cooking treatment, based on 15 determinations, and reported in Table 2. This statistical analysis is very useful for appraising what combinations meat product/cooking treatment lead to significant variation in the final concentration of nitrite and nitrate in cured meats. We hope this explanation can address the comment. Moreover, please note that a new figure (Figure 1) has been added, in order to simplify the reading of data reported in Table 1.

Reviewer 2 Report

Dear, considering the manuscript submitted for review, my opinion is that manuscript present valuable and useful results, for industry, regulatory bodies, and consumers. However, some revisions are needed.
In the section Introduction, legislation must be well described in terms of the maximum amount of nitrate or nitrite that may be added in processed meat expressed as NaNO2 or NaNO3, depending on the type of product.

Materials and Methods

Exposure Assessment and Risk Characterization must be separately explained. Likewise, this section is not well described, and thus must be improved.

Despite nitrite and nitrate are not carcinogenic substances, Agency on Research on Cancer (IARC) has classified processed meat as carcinogenic to humans (Group 1 carcinogen), based on sufficient evidence in humans that the consumption of processed meat causes colorectal cancer (CRC) and that red meat consumption is probably carcinogenic to humans (Group 2A). Therefore, exposure should be calculated for all the processed meat categories by the deterministic approach involving the average probable daily intake (APDI) method (JECFA—Joint FAO/WHO Expert Committee on Food Additives. Evaluation of Certain Contaminants in Food: Eighty-Third Report of the Joint FAO/WHO Expert Committee on Food Additives; WHO Technical Report Series 1002; WHO: Geneva, Switzerland, 2017;pp. 1–166. Available online: https://apps.who.int/iris/handle/10665/254893). Furthermore, to evaluate the adequacy of intakes/risk characterization, the calculated estimated daily intake (EDI) must be compared with the ADIs proposed by the EFSA (EFSA Panel on Food Additives and Nutrient Sources added to Food (EFSA ANS Panel); Mortensen, A.; Aguilar, F.; Crebelli, R.; Di Domenico, A.; Dusemund, B.; Frutos, M.J.; Galtier, P.; Gott, D.; Gundert-Remy, U.; et al. Scientific Opinion on the re-evaluation of potassium nitrite (E 249) and sodium nitrite (E 250) as food additives. EFSA J. 2017, 15, 4786).

Section Results In order to better visualize of results presented in Table 1. I suggest replacing table 1 with graphs, for each meat product (Similar to graph 4. but adjusted to meat products.

Please, the Margin of Safety (MoS) assessment replace according to the previous suggestion, using methods proposed by JECFA and EFSA.

Author Response

The authors would like to thank the reviewer for his effort in improving the scientific impact of the Paper. The manuscript has been revised, according to reviewer’s suggestions, editing corrections and rewording the text where necessary. Please note that the references to lines are referred to the pdf version of the revised article.

Reply to Reviewer 2

Dear, considering the manuscript submitted for review, my opinion is that manuscript present valuable and useful results, for industry, regulatory bodies, and consumers. However, some revisions are needed.

In the section Introduction, legislation must be well described in terms of the maximum amount of nitrate or nitrite that may be added in processed meat expressed as NaNO2 or NaNO3, depending on the type of product.

Response: As suggested by the reviewer, some examples of legal limits have been added, taking into account the most significant results obtained in this study (lines 40-43).

Materials and Methods

Exposure Assessment and Risk Characterization must be separately explained. Likewise, this section is not well described, and thus must be improved.

Despite nitrite and nitrate are not carcinogenic substances, Agency on Research on Cancer (IARC) has classified processed meat as carcinogenic to humans (Group 1 carcinogen), based on sufficient evidence in humans that the consumption of processed meat causes colorectal cancer (CRC) and that red meat consumption is probably carcinogenic to humans (Group 2A). Therefore, exposure should be calculated for all the processed meat categories by the deterministic approach involving the average probable daily intake (APDI) method (JECFA—Joint FAO/WHO Expert Committee on Food Additives. Evaluation of Certain Contaminants in Food: Eighty-Third Report of the Joint FAO/WHO Expert Committee on Food Additives; WHO Technical Report Series 1002; WHO: Geneva, Switzerland, 2017;pp. 1–166. Available online: https://apps.who.int/iris/handle/10665/254893). Furthermore, to evaluate the adequacy of intakes/risk characterization, the calculated estimated daily intake (EDI) must be compared with the ADIs proposed by the EFSA (EFSA Panel on Food Additives and Nutrient Sources added to Food (EFSA ANS Panel); Mortensen, A.; Aguilar, F.; Crebelli, R.; Di Domenico, A.; Dusemund, B.; Frutos, M.J.; Galtier, P.; Gott, D.; Gundert-Remy, U.; et al. Scientific Opinion on the re-evaluation of potassium nitrite (E 249) and sodium nitrite (E 250) as food additives. EFSA J. 2017, 15, 4786).

Please, the Margin of Safety (MoS) assessment replace according to the previous suggestion, using methods proposed by JECFA and EFSA. 

Response: Thanks for your comment. Please note that, as described at lines 168-169, the MoS is based on the ratio between ADI/estimated exposure dose, as suggested by the reviewer. Moreover, the reference parameters reported in the EFSA Scientific Opinion cited by the reviewer has been taken into consideration (ref. n.9, now 11) for all related calculations. In order to satisfy, as much as possible, the reviewer’s request, the percentages of ADI have been calculated and added in the revised version of table 3. Regarding the first part of the comment, please note that the focus of this study was not the evaluation of carcinogenicity of different meat products, but the effect of cooking on the residual level of nitrite and nitrate in meats. Indeed, the study does not provide original data on meat consumption. The exposure study, related to the intake of 2 additives was only added for obtaining a contribution to risk assessment. The authors are sorry, but they do not find other possible evaluations useful to improve the risk exposure study, given the not carcinogenicity of both nitrite and nitrate.

Section Results In order to better visualize of results presented in Table 1. I suggest replacing table 1 with graphs, for each meat product (Similar to graph 4. but adjusted to meat products.

Response: As suggested by the reviewer, data from table 1 have been re-elaborated to provide a new figure (Figure 1).

Round 2

Reviewer 2 Report

Dear, regarding the following manuscript, my opinion is that the authors significantly improved the manuscript according to the reviewer's suggestions. 

But, newly added references belong to the coauthor added in a revised version of the manuscript, making it inappropriate or unprofessional. 

Author Response

The authors would like to thank again the reviewer for his effort in improving the scientific impact of the Paper.

Reply to Reviewer 2

Dear, regarding the following manuscript, my opinion is that the authors significantly improved the manuscript according to the reviewer's suggestions.

But, newly added references belong to the coauthor added in a revised version of the manuscript, making it inappropriate or unprofessional.

Response: Thanks for your comment. The references were added due to their significance in the field, in order to complete the references section. However, following the reviewer’s remark, they were removed from the paper.
